# Co-Zeolitic Imidazolate Framework@Cellulose Aerogels from Sugarcane Bagasse for Activating Peroxymonosulfate to Degrade P-Nitrophenol

**DOI:** 10.3390/polym13050739

**Published:** 2021-02-27

**Authors:** Wen Sun, Kunyapat Thummavichai, Ding Chen, Yongxin Lei, Hui Pan, Taize Song, Nannan Wang, Yanqiu Zhu

**Affiliations:** Guangxi Institute Fullerene Technology (GIFT), Key Laboratory of New Processing Technology for Nonferrous Metals and Materials, Ministry of Education, School of Resources, Environment and Materials, Guangxi University, Nanning 530004, China; Sunwen0@outlook.com (W.S.); kt302@exeter.ac.uk (K.T.); 1914402004@st.gxu.edu.cn (D.C.); leiyongxin0@outlook.com (Y.L.); ph20003@163.com (H.P.); songtaize@sina.com (T.S.); y.zhu@gxu.edu.cn (Y.Z.)

**Keywords:** metal organic framework, bagasse cellulose aerogel, P-Nitrophenol, advanced oxidation process

## Abstract

An efficient, green and reusable catalyst for organic pollutant wastewater treatment has been a subject of intense research in recent decades due to the limitation of current technologies. Cellulose based aerogel composites are considered to be an especially promising candidate for next-generation catalytic material. This project was conducted in order to evaluate the behavior and ability of green and reusable sugarcane bagasse aerogels to remove P-Nitrophesnol from waste-water aqueous. Co-Zeolitic imidazolate framework@ sugarcane bagasse aerogels composite catalysts were successfully prepared via simple in situ synthesis. The structure of hybrid aerogels and their efficient catalyst in peroxymonosulfate (PMS) activation for the degradation of p-nitrophenol (PNP) was investigated. As a result, the hybrid aerogels/PMS system removed 98.5% of PNP (10 mg/L) within 60~70 min, while the traditional water treatment technology could not achieve this. In addition, through a free radical capture experiment and electron paramagnetic resonance (EPR), the degradation mechanism of PNP was investigated. Further research found that the hybrid aerogels can effectively activate PMS to produce sulfate (${\mathrm{SO}}_{4}^{•\;-}$) and hydroxyl (${\mathrm{OH}}^{•\;}$). Both of them contributed to the degradation of PNP, and ${\mathrm{SO}}_{4}^{•\;-}$ plays a crucial role in the degradative process. The most important feature of hybrid aerogels can be easily separated from the solution. The obtained results showed that the outer coating structure of cellulose can stabilize Co-ZIF and reduce the dissolution of cobalt ions under complex reaction conditions. Moreover, the prepared hybrid aerogels exhibit excellent reusability and are environmentally friendly with efficient catalytic efficiency. This work provides a new strategy for bagasse applications and material reusability.

## 1. Introduction

The volume of wastes produced by sugar and associated industries has increased year by year. It is reported that 166 million tons of bagasse waste were produced worldwide in 2020 [1], which causes huge pressure on landfills and environmental pollution due to their incineration activities. Therefore, bagasse has received global attention from researchers for various technologies, such as supercapacitor and adsorbent [2,3], due to their inherited properties. The comprehensive utilization of bagasse plays an important role for sugarcane enterprises to develop a circular economy, improve economic efficiency and achieve sustainable development for the sugar industry. For the last three decades, bagasse has been used for the biotransformation process to produce industrial products such as bioethanol, xylitol, and special enzymes [4,5]. However, these biotransformation applications have not been widely used due to their limitations or weaknesses in terms of technologies and economics [6,7]. Cellulose is the main component of bagasse, accounting for 54% of the total bagasse, and it is also the world’s most abundant natural, renewable and biodegradable polymer [8]. Cellulose and its derivatives have many essential application roles in the fiber, paper, film, polymer, coating and other industries [9], such as preparing naturally degradable straws to replace plastic straws [10] and lignocellulosic films with robust mechanical properties [11].

Environmental pollution has caused severe limitations to the development worldwide, and the ecological issues that come with it have also brought considerable challenges to the world. Toxic organic pollutants in the aquatic environment are receiving global attention due to their adverse impacts on human health and environmental ecosystems [12]. P-Nitrophenol (PNP) is a major ecological wastewater organic pollutant and is difficult to biodegrade. PNP is commonly found in industrial wastewater, insecticides, herbicides, explosives, synthetic dyes and pharmaceuticals [13,14]. What is more, it causes direct harm to aquatic organisms and pollutes groundwater through surface penetration, causing pollution of drinking water and domestic water. Acute exposure to PNP can cause anemia, skin and eye irritation, and damage the kidneys and liver [15,16]. In recent years, the pollution of PNP in the aquatic environment has become a significant concern due to the limitations of conventional wastewater treatment technology that is not effective enough to treat PNP pollutants [17,18]. Among all types of wastewater treatment technologies, advanced oxidation processes (AOPs) have proven to be a superior choice for the effective removal of organic compounds in water media [19,20]. AOPs based on sulfate radical (${\mathrm{SO}}_{4}^{•\;-}$) (SR-AOPS) has received increasing attention due to a series of advantages, such as ${\mathrm{SO}}_{4}^{•\;-}$ can possess an oxidation potential (2.5–3.1 V vs. normal hydrogen electrode (NHE)), which is even higher than ${\mathrm{OH}}^{•\;}$ (1.8–2.7 V vs. NHE) [21] and reacts more selectively and efficiently via electron transfer with organic compounds [22]. ${\mathrm{SO}}_{4}^{•\;-}$ can be produced by ultrasound, photolysis, pyrolysis or chemical activation of peroxymonosulfate (PMS) [22,23,24,25]. Oxone, a commercial name of potassium-based PMS (2KHSO_5_·KHSO_4_·K_2_SO_4_), is a versatile and environmentally friendly oxidant. It is widely used for bleaching, cleaning and disinfection, and more importantly as favorable source of PMS [26,27]. In fact, PMS can be activated by a variety of transition metal ions, such as Mn^2+^, Ce^3+^, Ni^2+^, Fe^2+^, where Co^2+^/PMS has been shown to have the most outstanding performance compared to traditional Fenton reactions with the same amount [28]. Zeolite, one of the most common molecular sieves that consists of a three-dimensional network of metal-oxygen tetrahedra. It has important properties such as size and shape selectivity of microporous structures, ion exchangeable sites, and good water stability [29,30]. These properties are beneficial for the active site of metal to better play the activation performance.

Cellulose aerogels have excellent physical properties such as high surface area, high porosity, biodegradability, recyclability, low cost and good water stability. Therefore, it is widely used in environmental treatment, energy storage and other aspects [31,32,33]. However, the research on the direct preparation of cellulose aerogel from bagasse is rarely reported. In this study, cellulose aerogel was prepared from bagasse materials and then was used as a substrate to grow the cobalt-based zeolite imidazole framework (Co-ZIF) material in-situ. The Co-ZIF@GEL as a heterogeneous catalyst effectively activates PMS oxidant for pollutant degradation. PNP is converted into harmless small molecules or directly mineralized into CO_2_ and H_2_O. Using aerogel as a skeletal structure to load Co-ZIF is very convenient for recovering and reusing Co-ZIF catalyst. This bagasse cellulose aerogel is a green and degradable material and easy to handle after use. The cellulose aerogel acts as a buffer and protection for the loaded Co-ZIF and, to a certain extent, alleviates the loss of transition metals. Simultaneously, Co-ZIF@GEL offers an excellent ability to degrade organic pollutants. The purpose of this research is to convert agricultural waste into a carrier for high-efficiency water pollution treatment materials, simplifying the recycling methods of materials without affecting the degradation performance, thereby ensuring the long-term stable use of materials.

## 2. Experimental

### 2.1. Materials and Reagents

The bagasse was purchased from Nanning Sugar Industry Co. Ltd. (Nanning, China) Cobalt nitrate hexahydrate (Co(NO_3_)_2_·6H_2_O, 99%), 2-methylimidazole (2-MeIM), methanol, sodium hydroxide (NaOH), urea, *N*,*N*′-methylene bisacrylamide (MBA), tert-butanol (TBA) and p-nitrophenol (PNP) were purchased from Sinopharm Chemical Reagent Co. Ltd. (Shanghai, China), 5,5-dimethyl-1-pyrroline N-oxide (DMPO) and potassium hydrogen persulfate (PMS, KHSO_5_·0.5 KSO_4_·0.5K_2_SO_4_) were ordered from Macklin Company (Shanghai, China). All materials used in this study were analytical grade without further purification. 

### 2.2. Preparation of ZIF-67

Co(NO_3_)_2_·6H_2_O (3 mmol) and 2-methylimidazole (24 mmol) was dissolved in 180 mL of methanol, respectively. After being stirred for 5 min at room temperature, the methanol solution of metal ions was poured into the 2-methylimidazole solution and under vigorous stirring for 12 hours. The mixed solution was allowed to stand for 12 h to make the suspension age. The resulting purple suspension was washed several times with methanol and the sample was collected by centrifugation.

### 2.3. Extraction of Cellulose from Bagasse

The sugarcane bagasse was first ground, washed, filtered and dried to remove unwanted residues or substances. The obtained bagasse powder was dewaxed in a Soxhlet extractor using a toluene-ethanol solution (toluene: ethanol = 2:1), heating at 90 °C for 6 h. Then, the pretreated bagasse was soaked in 600 mL of NaOH solution to lignin removal, heated at 80 °C and stirred. After extraction, purified bagasse cellulose was washed with deionized water and ethanol several times to remove residual impurities. Subsequently, the required amount of 10 g cellulose fibers were bleached in a NaClO_2_ (4 wt %) solution. The pH was adjusted with glacial acetic acid, heated and stirred to obtain a white cellulose suspension. The white suspension was collected, filtered and repeatedly washed with deionized water to acquire a neutral pH. The final products were dried in a vacuum drying oven at 60 °C for 12 h to obtain purified sugarcane bagasse cellulose.

### 2.4. Preparation of Cellulose Aerogel

Cellulose aerogel was prepared by the sol-gel method, in the intermediate processing, cellulose hydrogel was obtained by MBA chemical crosslinking [34]. The cellulose solution was a mixture of NaOH/urea/H_2_O with a mass ratio of 7:12:81 as described in literature [35]. 3 g of bagasse cellulose was added to 97 g of the prepared cellulose solution, and then mixed and froze in the refrigerator overnight at −12 °C. The frozen bagasse cellulose solution was vigorously stirred to obtain a relatively transparent liquid, 1 g of MBA was added to it, and stirring was continued for 30 min with mechanical agitation. The transparent solution was then poured into a well-plate and was allowed to stand overnight to obtain a cellulose hydrogel. The cellulose hydrogel was removed from the well plate and was washed with an aqueous ethanol solution to neutrality and residual chemicals were removed. Finally, the cellulose hydrogel was placed in a freeze dryer for 24 h to obtain aerogel cellulose.

### 2.5. Preparation of Co-ZIF@GEL Composite Aerogel

Co(NO_3_)_2_·6H_2_O(3 mmol) and 2-methylimidazole (24 mmol) were dissolved in 90 mL of ethanol solution, respectively (Named Solution A and B). The cellulose hydrogel was soaked in solution A for 12 h. After that, the solution A with hydrogel and solution B was uniformly mixed for another 12 h. The Co-ZIF@cellulose hydrogel was washed with ethanol and water to remove excess chemicals. Finally, the Co-ZIF@cellulose hybrid aerogel was obtained by freeze-drying.

### 2.6. Characterization 

The cellulose composition was analyzed using X-ray diffraction (XRD, Bruker D8, Berlin, Germany) with Cu Kα radiation (100 mA and 40 kV) at a scan rate of 5° min^−1^. The surface morphologies of MOFs loaded on the aerogel were studied using field emission scanning electron microscopy (FESEM, Hitachi S-4800, Tokyo, Japan). The hybrid cellulose aerogel was characterized by a thermos gravimetric analyzer (TGA, TG 209 F1, Germany) with a heating rate of 20 °C min^−1^ from 30 to 800 °C under nitrogen atmosphere. The electron paramagnetic resonance (EPR, Bruker A300, Berlin, Germany) was used to detect free radicals in the system. The instrument was operated with modulation amplitude 3G, a microwave power of 20.00 mW, a microwave frequency of 9.79 GHz and a modulation frequency of 100 kHz. After degradation experiments, the cobalt elements were analyzed by inductively coupled plasma atomic emission spectroscopy (ICPS-7510).

### 2.7. Catalysis Experiment Procedure and Analysis Method

The catalysis experiments were carried out in a 250 mL beaker at 25 °C. The PNP stock solution was dissolved in deionized water and stirred overnight for complete dissolution. The concentration of PNP was 10 mg/L and the initial pH was 6.8. A certain amount of catalyst is added to the PNP solution and stirred for 5 min to completely dispersing of the catalyst. Then, add a predetermined amount of PMS. The reaction solution (1 mL) was drawn out at regular intervals. The catalytic efficiency is C_t_/C_o_ (C_t_: PNP concentration at reaction t, C_o_: PNP initial concentration), and the PNP concentration is measured at 317 nm using an ultraviolet spectrophotometer. For all kinetic experiments under different factors, the degradation of PNP conforms to the pseudo-first-order kinetic equation (Equation (1)).
(1)$$-\ln C_{t}/{\;C}_{0}=K_{\mathrm{obs}}t$$
where C_0_ and C_t_ are the PNP concentration when the reaction time is 0 and t, respectively, and $K_{\mathrm{obs}}$ is the quasi-first-order constant(min^−1^). The loadings of hybrid aerogels were determined by a density difference calculation (Equation (2)).
(2)$$\mathrm{Loading}\;\mathrm{rate}\;=\;\left(\rho_{C}-\rho_{a}\right)/\rho_{c}$$

The $\rho_{C}$ represents the density of hybrid cellulose aerogel, $\rho_{a}$ represents the density of cellulose aerogel. The calculated loading rate of hybrid aerogel was 48 ± 1.8%.

## 3. Results and Discussions

### 3.1. Characterization of Co-ZIF@GEL

The production process of Co-ZIF@GEL is shown in Figure 1. Co-ZIF@GEL was prepared by the sol-gel method. First, the cellulose was extracted and purified from bagasse, and then the cellulose hydrogel was organized by the NaOH/Urea method. Then, Co-ZIF was grown in situ and was anchored on the hydrogel; a purple hydrogel was prepared. Finally, Co-ZIF@GEL was obtained by freeze-drying. It is obvious that the color of the hybrid aerogel shows vivid purple color due to the Co-ZIF. In the photograph (Figure 1), hybrid aerogel can stand on the tip of the leaf, and from Equation (2) it can be calculated that hybrid aerogels have an ultralow density of 15–22 mgcm^−3^. It can be seen that bagasse aerogel has large pores and thin pore walls, super large porosity and honeycomb structures with a diameter of up to 100 μm (Figure 2a,b). The thin pore wall can increase the contact area of water and aerogel. ZIF-67 presents a rhombic dodecahedron structure with a size of about 100nm (Figure 2c). The loading of Co-ZIF will not affect the porous structure of the cellulose aerogel (Figure 2d). Co-ZIF is evenly distributed on the pore surface of the composite aerogel and were wrapped by a thin layer of cellulose as a protection film (Figure 2e,f). Due to the hydrophilicity of extracted cellulose fibrils used for the aerogel’s unique structure preparation, a significant negative impact on degradation rate of PNP was not recorded. The loaded Co-ZIF was compared with ZIF-67. The original rhombic dodecahedron structure has become a spherical structure with burrs, as shown in Figure 2f. This is because the Co-ZIF particles grow in-situ on the aerogel changed the crystal growth method, resulting in a change in the structure.

Figure 3a shows the XRD pattern of bagasse, cellulose after the purification process, and a final aerogel product. The diffraction peaks of bagasse were noticed at 2θ = 14.6°, 16.5°, and 22.5°, and belonged to cellulose type I. After the purification process, the transformation of cellulose I to cellulose II of the extracted cellulose sample appears, as evidenced by the appearance of diffraction peaks at 2θ = 12.3°, 20.1° and 21.9° [36]. The dispersion peak of cellulose was between 20–30° and the characteristic peak (2θ = 7.3°,10.3°,12.7°) of ZIF-67 on the XRD pattern of Co-ZIF@GEL composites confirmed that Co-ZIF is successfully loaded on the GEL structure. The Co-ZIF characteristic peak in the Co-ZIF@GEL composite is not obvious compared to pristine Co-ZIF materials. This may be due to Co-ZIF being encapsulated in composite aerogel. 

TG analysis (30–800 °C) was conducted to investigate the thermal performance of as-prepared GEL, Co-ZIF@GEL, and ZIF-67 samples (Figure 3b). The thermal stability of ZIF-67 occurs up to 550 °C. The result shows that about 9% of the GEL and Co-ZIF@GEL weight was lost in the range of 30–120 °C, for samples caused by the water decomposition, as the temperature rises. The composite decomposes gradually; about 60% over the temperature range of 220–380 °C, which is assigned to a cellulose decomposition and carbon formation. A slight drop of GEL and weight of Co-ZIF@GEL at a range of 600–800 °C is due to the residual coke after burning. The maximum decomposition temperature of the GEL after Co-ZIF introduction increased from about 350 to 400 °C, showing that the addition of Co-ZIF slightly improves the thermal stability of the GEL. Hence, the thermal stability of Co-ZIF@GEL was improved. After MBA crosslinking and in-situ growth of ZIF-67, the chemical structure changes of the metal-organic framework/cellulose nanocomposite are shown in Appendix A. In the Fourier-transform infrared (FT-IR) spectrum, the characteristic peaks at 3418 cm^−1^, 2871 cm^−1^, 1651 cm^−1^, 887 cm^−1^ are all the typical bands of cellulose molecules [34]. The characteristic peaks at 1421 cm^−1^, 1327 cm^−1^, 688 cm^−1^ are all the typical bands of ZIF-67 molecules. The absorption peak appeared at 1537 cm^−1^, which is the N–H bond bending vibration absorption peak after MBA successfully cross-linked cellulose molecules. Obviously, hybrid aerogels retain the characteristic peaks of both ZIF-67 and cellulose. 

### 3.2. Catalytic Degradation Performance of Composite Aerogel

The result of composite aerogel catalytic degradation ability on PNP is shown in Figure 4a. There is almost no sign of change in pollutants’ concentration when only PMS is added to the solution, indicating that PMS is an efficient green oxidant. In the absence of a catalyst, there is almost no degradation performance. Furthermore, the adsorption capacity of Co-ZIF@GEL and ZIF-67 for PNP is relatively limited. The adsorption effect is far less potent than that of the composite aerogel/PMS system for PNP degradation. In order to intuitively reflect the effect of cellulose inclusion on the catalytic active site, the corresponding $K_{\mathrm{obs}}$ is shown in Figure 4b; the $K_{\mathrm{obs}}$ dropped from 3.15 × 10^−3^ to 8.05 × 10^−4^ (Figure 4b) indicating that the cellulose coating has a slight negative effect on the degradation ability of Co-ZIF@GEL.

#### 3.2.1. The Effect of Catalyst Dosage and PMS Concentration

The result of catalyst dosage on PNP degradation is presented in Figure 5a. It is worth mentioning that when PMS exists alone, the degradation efficiency of PNP is quite limited and the degradation rate is less than 4%. This could be due to the individual PMS not being able to generate free radicals. The degradation efficiency of PNP significantly improves upon the addition of Co-ZIF@GEL into the PMS system. When the dosage of Co-ZIF@GEL catalyst was 60 mg/L, PNP degradation experiments have proved that about 90.7% of the PNP can be removed within 60 min. The degradation of PNP increases rapidly between 60–100 mg/L of catalyst usage, indicating that the reaction with the composite aerogel surface is the reason for limiting the degradation rate. When the composite aerogel usage continues to increase to 140 mg/L, the degradation rate of PNP did not increase significantly because the concentration of PMS had become the main factor affecting the degradation rate. Moreover, when free radicals are in a high concentration state, self-quenching may occur (Equations (3) and (4)), which also inhibits the continuous increase of ${\mathrm{SO}}_{4}^{•\;-}$.
(3)$${\mathrm{SO}}_{4}^{•\;-}\;+\;{\mathrm{SO}}_{4}^{•\;-}\;\to \;S_{2}O_{8}^{2\;-}$$
(4)$${\mathrm{SO}}_{4}^{•\;-}\;+\;{\mathrm{OH}}^{•\;}\;\to \;{\mathrm{HSO}}_{5}^{-}$$

The effect of PMS concentration on PNP degradation is shown in Figure 5b. The results show that when the PMS concentration increases from 0.7 mM to 1 mM, it significantly impacts the catalytic performance, leading to an increased degradation rate from 90% to 98% by 70 min, whereas the degradation rate remained nearly constant at PMS concentrations when the PMS dosage increased from 1.0 mM to 1.3 mM. At low PMS concentrations, the increased rate can be attributed to the increased adsorbed PMS on the catalyst’s surface with increasing PMS concentration. Hence, the catalytic performance does not fulfill at 0.7 mM of PMS due to the unsaturation of PMS, so it cannot generate enough free radicals to degrade PNP. At PMS concentrations above 1 mM, the saturated PMS adsorption was reached and the maximum catalytic performance was obtained, which resulted in nearly identical reaction rates even with further increasing the initial PMS concentration. When the amount of PMS added reaches 1 mM, efficient removal of PNP can be achieved. Considering the price of PMS, increasing the PMS concentration for catalysis is not an economic strategy. Therefore, it is determined that 1 mM PMS is suitable for the catalytic degradation process under the same environmental conditions.

#### 3.2.2. Influence of Initial pH and Dissolution of Cobalt Ion after the Reaction

The initial pH of a solution is an essential factor affecting the catalytic performance of the Co-ZIF@GEL/PMS system. The effect of different initial pH values (3, 7, 9) on the catalytic performance was further evaluated and the results are shown in Figure 5c. The results show that the degradation of PNP was pH-dependent. Almost all PNP can be degraded in the Co-ZIF@GEL/PMS system within 70 min at a neutral pH (6.8). Meanwhile, the degradation efficiency is significantly reduced in acidic conditions. This may be due to the CoOH^+^ formation of the catalyst (Equation (5)), Therefore, the production of ${\mathrm{SO}}_{4}^{•\;-}$ is suppressed. Excessive $H^{+}$ may hinder the further transformation of ${\mathrm{HSO}}_{5}^{-}$. There is no noticeable effect on the degradation efficiency of PNP pH = 9 conditions. The results demonstrated that the Co-ZIF@GEL/PMS system has excellent resistance under alkaline conditions, whereas the degradation rate is slightly better than pH 6.8 levels. The relatively fast degradation rate under alkaline conditions could be due to ${\mathrm{OH}}^{-}$, which helps to promote the conversion of ${\mathrm{Co}}^{3+}$ to ${\mathrm{Co}}^{2+}$ (Equation (8)) quickly.
(5)$${\mathrm{Co}}^{2+}+H_{2}O\to {\mathrm{CoOH}}^{+}+H^{+}$$

The degraded solution was taken out and the concentration of cobalt ions in the solution was measured using ICP. The dissolution of cobalt ions after the active reaction of ZIF-67 and Co-ZIF@GEL is shown in Figure 5d. Without the GEL package, the cobalt ion concentration reached 8.6 mg/L. After the composite aerogel reaction, the cobalt ion concentration showed a significant decrease, indicating that the cellulose has protected the transition metal cobalt and anchoring effect. 

### 3.3. Identification of Reactive Species and Activation Mechanisms

In order to understand the degradation mechanism, the experimental results from using methanol and tert-butanol (TBA) as free radical scavengers on PNP degradation in the Co-ZIF@GEL/PMS system are shown in Figure 6a. Studying the inhibitory effect of scavengers on reflections can identify the oxidizing radical species in a Co-ZIF@GEL/PMS system. Methanol is regarded as a scavenger of both ${\mathrm{SO}}_{4}^{•\;-}$ and ${\mathrm{OH}}^{•\;}$, while TBA is considered the scavenger of only ${\mathrm{OH}}^{•\;}$ [37]. PNP can be almost completely removed within 70 min when there is no scavenger. The degradation of PNP is slightly inhibited at 100 mM TBA concentration, suggesting that the contribution of ${\mathrm{OH}}^{•\;}$ to PNP degradation is very limited. Meanwhile, 100 mM of methanol inhibited the oxidation reaction with only 15% of PNP being degraded in the oxidation, indicating that ${\mathrm{SO}}_{4}^{•\;-}$ is the dominant radical species in the Co-ZIF@GEL/PMS system at pH 6.8.

For further investigation of the active species involved in the Co-ZIF@GEL/PMS system, electron paramagnetic resonance (EPR) was conducted with DMPO as a spin trap for ${\mathrm{SO}}_{4}^{•\;-}$ and ${\mathrm{OH}}^{•\;}$. EPR could effectively detect transient free radicals, ${\mathrm{SO}}_{4}^{•\;-}$ and ${\mathrm{OH}}^{•\;}$ at the same time in the process of PMS activation [38]. As shown in Figure 6b, there are two different free radicals species present in the system. Further investigation reveals that the intensity ratio of the characteristic peak is 1:2:2:1 (αH = αN = 14.8 G), corresponding to the signals of ${\mathrm{OH}}^{•\;}$, while the signals of ${\mathrm{SO}}_{4}^{•\;-}$ exist around the peaks of ${\mathrm{OH}}^{•\;}$ (αN = 13.2 G, αH = 9.6 G, αH = 1.48 G, αN = 0.78 G) [39,40]. These results were consistent with the results of free radical quenching experiments in Figure 6a. In other words, with the addition of hybrid aerogels, the Co-ZIF@GEL/PMS system can produce ${\mathrm{SO}}_{4}^{•\;-}$ and ${\mathrm{OH}}^{•\;}$, which eventually leads to the degradation of PNP [41].

X-ray photoelectron spectroscopy (XPS) was used to determine the element valence of the synthesized Co-ZIF. As shown in Figure 7a, the XPS spectrum shows the characteristic peaks of C1s, N1s, O1s, and Co2p, indicating the existence of these four elements. The change of peak intensity before and after catalytic use is mainly reflected in Co2p. Figure 7b shows the XPS spectra of Cobalt before and after catalytic use. The two main peaks located at 780.2 and 781.5eV reflect Co^3+^ 2p_3/2_ and Co^2+^ 2p_3/2,_ respectively. Co2+ 2p1/2 contributes to the other peak at 795.8 eV_,_ and the satellite feature peak of 786.6 and 802.6 eV illustrates the presence of Co^2+^ oxides species. The XPS results show that there is Co^2+^ and Co^3+^ in the composite material. The Co^3+^/Co^2+^ ratio of Co-ZIF increased from 0.38 to 2.33 after use, indicating that the valence of Cobalt increased from +2 to +3 by transferring electrons to PMS to generate ${\mathrm{SO}}_{4}^{•\;-}$ and ${\mathrm{OH}}^{•\;}$.

In summary, the above results demonstrate that the Co-ZIF@GEL has a potential application for PNP pollution degradation by activation of PMS. ${\mathrm{SO}}_{4}^{•\;-}$ and ${\mathrm{OH}}^{•\;}$ radicals proved to be the key active species for PNP degradation in the oxidation reaction system. As shown in Appendix A, the UV absorption peak appeared at 317nm and the peak intensity decreased with the increase of time, while no other UV absorption peak appeared, indicating that PNP is not converted into other organic pollutants but into harmless inorganic substances ^36^. PMS can be activated by ${\mathrm{Co}}^{2+}$ in the Co-ZIF@GEL structure to produce ${\mathrm{SO}}_{4}^{•\;-}$ and ${\mathrm{OH}}^{•\;}$. When the cobalt ions in ZIFs are adsorbed and desorbed by external molecules in the water, the cobalt ions can be exchanged between ${\mathrm{Co}}^{3+}$ and ${\mathrm{Co}}^{2+}$ [42]. Therefore, we can assume that the activation of PMS by cobalt ion produces ${\mathrm{SO}}_{4}^{•\;-}$ and ${\mathrm{OH}}^{•\;}$ according to the following reaction equation (Equations (6)–(9)):(6)$${\mathrm{Co}}^{2+}+{\mathrm{HSO}}_{5}^{-}\;\to {\mathrm{Co}}^{3+}+{\mathrm{SO}}_{4}^{•\;-}+{\mathrm{OH}}^{•\;}$$
(7)$${\mathrm{Co}}^{3+}+e^{-}\to {\mathrm{Co}}^{2+}$$
(8)$${\mathrm{SO}}_{4}^{•\;-}+{\mathrm{OH}}^{-}\;\to {\mathrm{OH}}^{•\;}+{\mathrm{SO}}_{4}^{2\;-}$$
(9)$${\mathrm{Co}}^{3+}+{\mathrm{HSO}}_{5}^{-}\;\to {\mathrm{Co}}^{2+}+{\mathrm{SO}}_{5}^{•\;-}+H^{+}$$

### 3.4. Stability and Practical Application of Co-ZIF@GEL

The hybrid aerogel was compressed entirely and the water was compressed (Figure 8). When the hand is released, the hybrid aerogel begins to absorb water and completely restores its original shape. The hybrid aerogel can be quickly taken out of the water, the water squeezed out and then recycled without changing the shape of the aerogel, which proves the excellent reusability of the aerogel. Although ZIFs have shown their potential as PMS activated heterogeneous catalysts, it is crucial to check their recyclability for long-term use of catalysts. To evaluate the reusability and stability of Co-ZIF@GEL in PMS activation, three consecutive PNP degradation experiments were carried out under the same reaction conditions (Figure 8e). Squeeze the used Co-ZIF@GEL with deionized water and wash it, put it into the reaction solution under the same experimental conditions and conduct three consecutive parallel experiments. The removal rates of PNP are 98.5%, 93.3%, and 53.1, respectively. It proves that Co-ZIF@GEL can show good activation ability for a long time. The decrease in removal rate may be due to long-term immersion in the PMS environment, resulting in the loss of transition metal elements and the coverage of pollutants on the active site. However, the adsorption capacity(q_e_) of magnetic activated carbon prepared from a plant by-product (corn husk) is 0.344 mg/g for PNP, which requires a lot of activated carbon [43]. Due to its good stability and reusability, Zn/Co-ZIF@GEL can handle water pollution treatment problems.

## 4. Conclusions

The Co-ZIF@GEL composite was successfully prepared using a simple in situ synthesis method and was used as an activator for PMS on PNP degradation. A green, cost-effective and highly flexible aerogel has been successfully converted from sugarcane bagasse and is effectively used as catalyst supports. Co-ZIF@GEL can effectively degrade organic pollutants as a catalyst for activating PMS. The Co-ZIF@GEL/PMS system revealed remarkable catalytic activity; almost all PNP (98.5%) was degraded in 70 min. This study shows that ${\mathrm{SO}}_{4}^{•\;-}$ is the dominating active species at pH 6.8 in the Co-ZIF@GEL/PMS system. Cobalt species were involved in the formation of both ${\mathrm{SO}}_{4}^{•\;-}$ and ${\mathrm{OH}}^{•}$ radicals. The cycle test results show that Co-ZIF@GEL has excellent recycling performance and, even after three cycles, it still maintains good degradation performance. Therefore, the Co-ZIF@GEL/PMS system has broad application prospects for PNP wastewater treatment. 

## Figures and Tables

**Figure 1 polymers-13-00739-f001:**
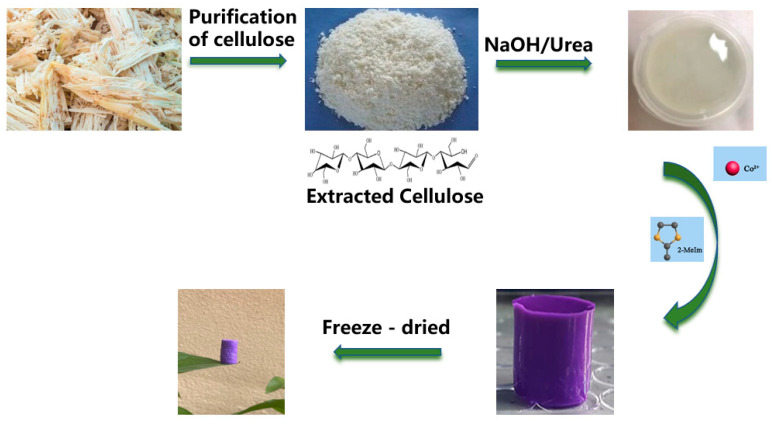
Schematic illustration of the fabrication process of Co-ZIF@GEL.

**Figure 2 polymers-13-00739-f002:**
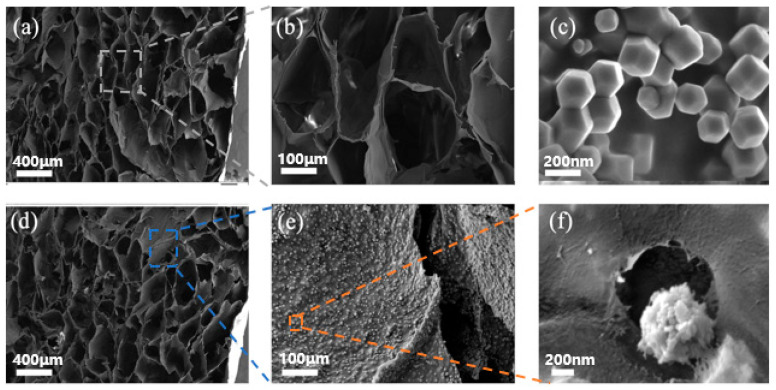
Scanning electron microscopy (SEM) image of (**a**,**b**) GEL; (**c**) ZIF-67; (**d**–**f**) Co-ZIF@GEL.

**Figure 3 polymers-13-00739-f003:**
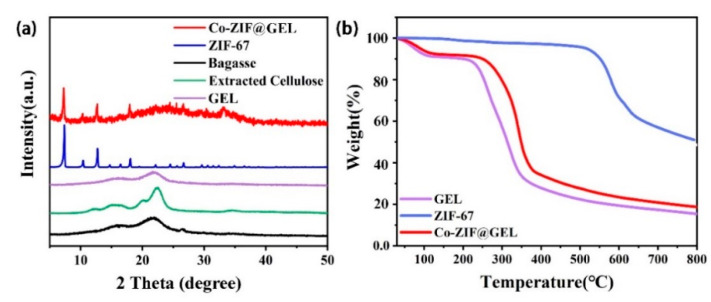
X-ray powder diffractograms of (**a**) Extracted cellulose from bagasse, ZIF-67, GEL and Co-ZIF@GEL; (**b**) TG curves of ZIF-67 and Co-ZIF@GEL.

**Figure 4 polymers-13-00739-f004:**
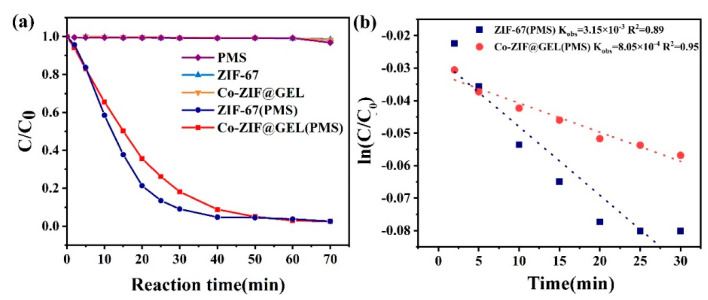
ZIF-67, Co-ZIF@GEL, PNP degradation performance under the peroxymonosulfate (PMS) system. (**a**) The result of composite aerogel catalytic degradation ability on PNP; (**b**) the effect of cellulose inclusion on the catalytic active site. (conditions: PNP = 10 mg/L, PMS = 1mM, ZIF-67 = 50 mg/L, Co-ZIF@GEL = 100 mg/L, GEL = 100 mg/L, PH = 6.8, T = 25 °C).

**Figure 5 polymers-13-00739-f005:**
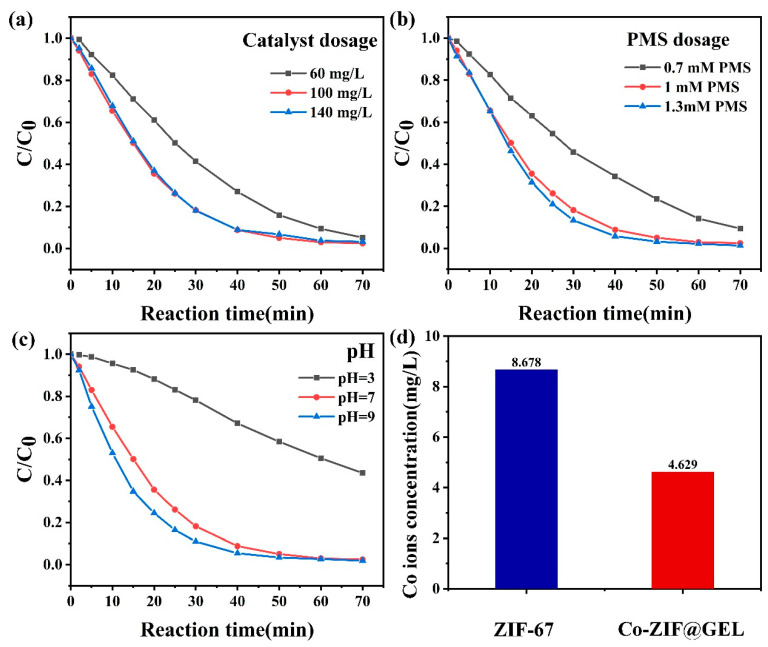
(**a**) The amount of catalyst (PMS = 1 mM) (**b**) The influence of PMS concentration (catalyst = 100 mg/L) on the degradation of PNP (**c**) The influence of initial pH on the reaction (**d**) The dissolution of cobalt ion after the reaction Contrast (conditions: ZIF-67 = 50 mg/L, Co-ZIF@GEL = 100 mg/L, PMS = 1 mM, PNP = 10 mg/L, T = 25 °C;).

**Figure 6 polymers-13-00739-f006:**
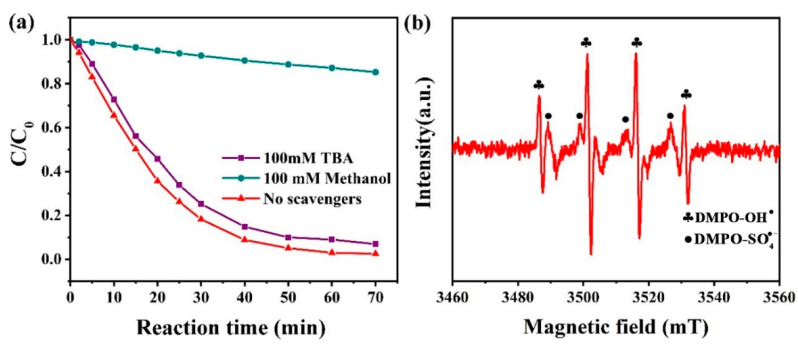
(**a**) The effect of different free radical scavengers on PNP degradation in Co-ZIF@GEL/PMS system (**b**) EPR spectrum of PMS activated by Co-ZIF@GEL within one minute (condition: Co-ZIF@GEL = 100 mg/L, PMS = 1 mM, PNP = 10 mg/L, T = 25 °C;).

**Figure 7 polymers-13-00739-f007:**
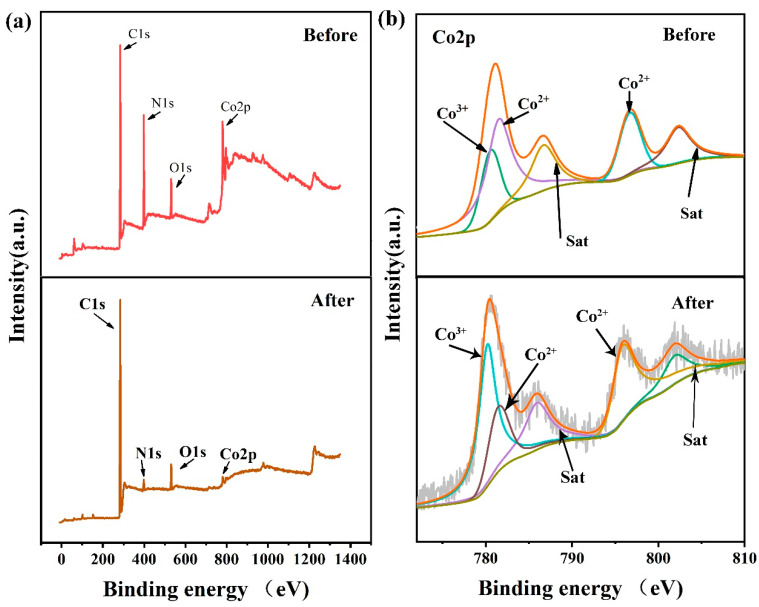
X-ray photoelectron spectroscopy (XPS) spectra of the Co-ZIFs particles: (**a**) full-range scan of the samples, (**b**) Co2p core level.

**Figure 8 polymers-13-00739-f008:**
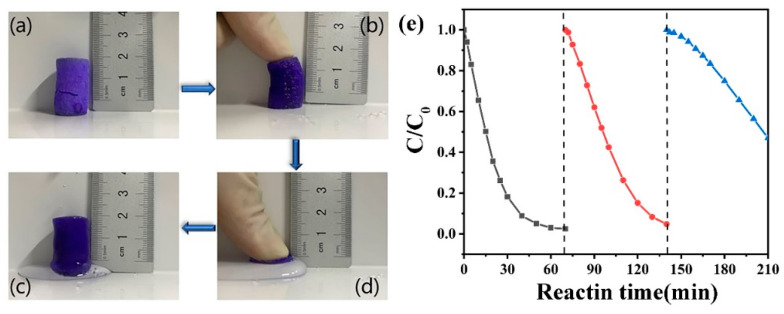
(**a**–**d**) show the process of restoring the water-containing hybrid aerogel to its original state after being compressed. (**e**) Recyclability of Co-ZIF@GEL for PNP removal.

## Data Availability

The data presented in this study are available on request from the corresponding author.

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
