# Peer review of "Co-Zeolitic Imidazolate Framework@Cellulose Aerogels from Sugarcane Bagasse for Activating Peroxymonosulfate to Degrade P-Nitrophenol"

_polymers, 2021, doi:10.3390/polym13050739_

Round 1

Reviewer 1 Report

The manuscript entitled "Co-Zeolitic imidazolate framework@cellulose aerogels from sugarcane bagasse for activating peroxymonosulfate to degrade p-Nitrophenol" described study on application of cellulose based composite for degradation of water pollutant. The presented study has moderate novelty impact  for polymer chemistry. However the manuscript could be published in Polymers Journal after significant modification according to comments and suggestions listed above.

Comments#

  1. Authors wrote in abstract; The hybrid aerogel/PMS system can remove almost all PNP within 60~70 min, while the ……….. The initial concentration of PNP should be added.

  1. Figure 1, should be rather placed in material and methods section not in results.
  2. The English should be revised and methods description should be changed to past simple passive form especially in sub section 2.4. Preparation of cellulose aerogel.
  3. “The resulting hybrid aerogel is so light that it can be placed on the leaf's tip without bending the leaf”. This kind of results description should be rather avoided and more scientific kind of description of results should be applied.
  4. “The water flow can quickly and massively pass be-tween the pores”….. Does some results in manuscript show the hydrodynamic properties of water in prepared composite? If not this phrase is only speculation and supposition.
  5. “the subsequent degradation experiment results show no significant negative impact”. Could authors precise on which properties of prepared material experiment was not showed negative impact ?
  6. Figure 3. The XRD images should be changed on … X-ray powder diffractograms
  7. Authors wrote that preparation process influence on crystallinity of bagasse cellulose. Why any crystallinity index was not calculated in base of XRD diffractograms ? How preparation of composite influence on cellulose microfibrils crystallites size ?
  8. Why only XRD and TG technique was used for composite characterization ? The FTIR or RAMAN technique should be also used for full characterization of composite structure and chemical composition.
  9. Could authors explain role of MBA in GEL preparation process? And why it is important for composite stabilization?
  10. TG analysis was conducted to investigate the thermal performance… This technique is used in polymer chemistry for investigation of influence of structural properties material according to their thermal degradation profile (stability) in specific conditions (atmosphere). Result should be also supplemented with specific values such as onset temperature, etc.
  11. ……o-ZIF could help to block heat and protect the GEL matrix to some extent. What Authors means with “block heat” ?
  12. The results section should be improved by more deep discussion of obtained results with comparison to actual knowledge of use cellulose based composites in PNP or other organic kind pollutants degradation e.g;

Ashour, Radwa M et al. “Green Synthesis of Metal-Organic Framework Bacterial Cellulose Nanocomposites for Separation Applications.” Polymers vol. 12,5 1104. 13 May. 2020, doi:10.3390/polym12051104

Y Wu, W Ren, Y Li, J Gao et al. Zeolitic Imidazolate Framework-67@Cellulose aerogel for rapid and efficient degradation of organic pollutants.

  1. The efficiency of degradation of PNP was analyzed according to changes in absorbance at 317 nm. Authors should also check the final products of this reaction by GCMS or LCMS/MS method for analysis that obtained product of reaction aren’t potentially more toxic than PNP. The cytotoxicity or phytotoxicity analysis could give also answer on this very important question.

  1. English and style of manuscript should be carefully revisited by English native speaker. In actual form the manuscript contains a lot of grammar inaccurateness that significantly reduce quality of reported research

Reviewer 2 Report

The manuscript entitled “Co-Zeolitic imidazolate framework@cellulose aerogels from sugarcane bagasse for activating peroxymonosulfate to degrade p-Nitrophenol” prepared by Sun and coworkers reports the preparation of Co-ZIF supported on cellulose aerogel for activating peroxymonosulfate to degrade p-Nitrophenol (PNP) a major ecological wastewater pollutant. The authors state that anchoring the Co-ZIF prevented the Co leaching and resulted in a recyclable catalyst that can degrade PNP within 60-70 min.  The authors tested various conditions that affect the degradation process including the catalyst dosage, PMS concentration, and pH.

Starting from section 3.2 (last paragraph on page 6), the manuscript becomes clear and well written. However, the first 6 pages include several issues. Additionally, thorough English proofreading of these pages is required. I suggest accepting this manuscript after English proofreading and correcting the following issue.

Introduction:

The authors should develop the introduction further and discuss the idea and sope of their work versus the state of the art. Additionally, they should explain to the readers the reasons behind their choice of materials. Why ZIF? Why Co?

Experimental

Section 2.4. Preparation of cellulose aerogel: the paragraph starts with:Co-ZIF@GEL is prepared by the sol-gel method.” meanwhile this section is devoted to the preparation of the aerogel. The first sentence is misleading in its current position and should be removed

Section 2.5:

  • The term cellulose hydrogen is repeated two times. This should be rectified if it is meant to be cellulose hydrogel Otherwise, explain what does cellulose hydrogen mean.
  • Does “the as-prepared Co(NO3)2·6H2O alcohol aqueous solution for 12h” mean Co-ZIF or Co-ZIF precursor? Replace the sentence with the correct term.
  • Indicate what is the composition of the “aqueous solution” used in washing.
  • What does the term neutrality” mean?

Discussion

With the exception of figure 5, all the figures appear before their first citation in the text.

Reviewer 3 Report

The manuscript is excellent. He pointed out the significant synthesis and put emphasis on inorganic polymerization.
The authors explained the mechanism flawlessly. I suggest printing it in your Journal. 

Author Response

Thank you for your review of the article and we will continue to work on it.

Round 2

Reviewer 1 Report

Dear Authors, manuscript was significantly improved. However some elements still a slight correction

  1. The abstract section, that is the most important part of manuscripts should be improved. I suggest split it to for clear parts like aim, results, conclusion. In present form it is hard to understand the main reason of conducted study.
  2. In introduction section the insertion about PMS was introduced without clear background. Please rephrase it slightly to be more smoothly.
  3. “Due to the hydrophilicity of cellulose and aerogel's unique structure, the subsequent degradation experiment results show no significant negative impact on degradation effect and degradation rate of PNP.” Sentence should be corrected eg;

“Due to the hydrophilicity of extracted cellulose fibrils used for  aerogel's unique structure preparation, significant negative impact on degradation rate of PNP was not recorded.”

  1. The style of manuscript writing was improved however it still need correction in many places. I recommend  ask the experienced researcher for help in polishing of final form of manuscript.
